# In Vitro Predictive Model for Intestinal Lymphatic Uptake: Exploration of Additional Enhancers and Inhibitors

**DOI:** 10.3390/pharmaceutics16060768

**Published:** 2024-06-05

**Authors:** Malaz Yousef, Conor O’Croinin, Tyson S. Le, Chulhun Park, Jieyu Zuo, Nadia Bou Chacra, Neal M. Davies, Raimar Löbenberg

**Affiliations:** 1Faculty of Pharmacy and Pharmaceutical Sciences, University of Alberta, Edmonton, AB T6G 2E1, Canada; malaz@ualberta.ca (M.Y.); ocroinin@ualberta.ca (C.O.); tsle@ualberta.ca (T.S.L.); zjieyu@ualberta.ca (J.Z.); 2Faculty of Pharmaceutical Sciences, University of Sao Paulo, Sao Paulo 05508-000, Brazil; chacra@usp.br; 3College of Pharmacy, Jeju National University, Jeju 63243, Republic of Korea; chpark@jejunu.ac.kr

**Keywords:** lymphatic uptake, chylomicrons, in vitro, enhancers, inhibitors

## Abstract

Drug absorption via chylomicrons holds significant implications for both pharmacokinetics and pharmacodynamics. However, a mechanistic understanding of predicting in vivo intestinal lymphatic uptake remains largely unexplored. This study aimed to delve into the intestinal lymphatic uptake of drugs, investigating both enhancement and inhibition using various excipients through our previously established in vitro model. It also examined the applicability of the model by assessing the lymphatic uptake enhancement of a lymphotropic formulation with linoleoyl polyoxyl-6 glycerides using the same model. The model successfully differentiated among olive, sesame, and peanut oils in terms of lymphatic uptake. However, it did not distinguish between oils containing long-chain fatty acids and coconut oil. Coconut oil, known for its abundance of medium-chain fatty acids, outperformed other oils. This heightened uptake was attributed to the superior emulsification of this oil in artificial chylomicron media due to its high content of medium-chain fatty acids. Additionally, the enhanced uptake of the tested formulation with linoleoyl polyoxyl-6 glycerides underscored the practical applicability of this model in formulation optimization. Moreover, data suggested that increasing the zeta potential of Intralipid^®^ using sodium lauryl sulfate (SLS) and decreasing it using (+/−) chloroquine led to enhanced and reduced uptake in the in vitro model, respectively. These findings indicate the potential influence of the zeta potential on intestinal lymphatic uptake in this model, though further research is needed to explore the possible translation of this mechanism in vivo.

## 1. Introduction

Intestinal lymphatic drug transport has recently garnered attention owing to the many potential benefits it presents for drug delivery [1,2]. Following absorption, some drugs pass across the intestinal enterocytes, and during this transit, these drugs associate with the excretory enterocyte lipoproteins chylomicrons [3]. This process underscores the potential and significance of exploiting intestinal lymphatic transport for drug delivery purposes.

Chylomicrons are spherical particles that are composed mainly of triglycerides (85–90%) in addition to phospholipids (7–9%), cholesterol and cholesteryl esters (3–5 and 1–3%, respectively), and apolipoproteins (1–2%) [4]. They principally play a role in absorbing and facilitating the systemic distribution of dietary fats and lipophilic vitamins [5]. Following digestion, when dietary triglycerides transform into free fatty acids and monoglycerides, a subsequent process of re-esterification occurs inside enterocytes. During this phase, the resulting triglycerides are encapsulated within chylomicrons, which serve as transportation carriers in the bloodstream through the lymphatic network [2,6].

In the context of pharmaceutical applications, specifically lymphatic-targeting—or lymphotropic—drugs, these enterocyte-formed chylomicrons offer a unique avenue. By ‘hitchhiking’ on these carriers, candidate molecules gain entry into circulation. Using chylomicrons as an approach holds the promise of evading the initial hepatic metabolism, commonly known as the first-pass effect, thereby elevating their bioavailability [5,7,8]. Alternatively, these drugs could accumulate within the lymphatic system, reaching increased concentrations at lymph node target sites. This concentration enhancement may translate into a more potent therapeutic impact with reduced off-target toxicity. This aspect is particularly important for compounds with immunomodulatory or anticancer properties, where maximizing their effect within the lymphatic system proves crucial [5,7].

In a previous study, we presented an in vitro model crafted to predict, inhibit, and enhance lymphatic uptake. Its foundation lies in the interaction of drugs with chylomicrons, a process documented for its ability to predict intestinal lymphatic uptake [9,10]. This model consists of two compartments: a donor compartment containing the drug solution under investigation and a receiver compartment filled with an artificial chylomicron medium (Intralipid^®^) [9]. These artificial chylomicrons serve as carriers for the drug molecules and mimic the behavior of naturally occurring chylomicrons in the body. To simulate the in vivo chylomicron-blocking effect and suppress drug release in an in vitro setting, pluronic L-81 (PL-81) was utilized. This chylomicron-blocking agent, which has been proven effective in both in vivo and Caco-2 cell culture models, demonstrated an inhibitory effect within the in vitro model [11,12,13]. Moreover, to enhance drug release into the receiver compartment and mimic lymphatic enhancement, peanut oil was used. This choice stemmed from peanut oil being contributes to the formation of chylomicrons and was guided by its potential to function as a carrier, facilitating increased drug entry into the receiver compartment [9,14]. 

In this study, the aim was to investigate other agents that would enhance or inhibit intestinal lymphatic uptake through the chylomicron pathway. Rifampicin served as the model drug in this study, consistent with the earlier investigation. Additionally, quercetin was used as a second xenobiotic to provide further confirmation in some experiments. Additional oils were explored to investigate their impact on enhancing intestinal lymphatic uptake. Olive, sesame, and coconut oils were chosen due to their varying percentages and chain lengths of different fatty acids, which are recognized for their impact on in vivo lymphatic uptake [4,15]. In order to deliver drugs through intestinal lymphatics, various formulation excipients and drug delivery systems have been and are being developed [1,7]. One example of the excipients used is Labrafil^®^. It consists of mono-, di-, and triglycerides and PEG6 (MW 300) mono- and diesters of linoleic (C18:2) acid. It is a non-ionic water-dispersible surfactant for lipid-based formulations to solubilize and increase the oral bioavailability of poorly water-soluble APIs [16]. A novel formulation of cannflavins was examined in this model system, with Labrafil^®^ M 2125 CS acting as the enhancer. Moreover, this study delved into the impact of the zeta potential on either enhancing or inhibiting intestinal lymphatic uptake. For this purpose, racemic chloroquine (C_18_H_26_ClN_3_) and sodium lauryl sulfate (C_12_H_25_NaO_4_S) were utilized. Chloroquine is an antimalarial drug that has been shown to reduce plasma levels of triglycerides and cholesterol [17]. At a physiological pH, chloroquine carries a positive charge. The purpose was to investigate whether this positive charge could influence its interaction with chylomicrons, consequently reducing triglyceride levels and potentially drug transportation through chylomicrons. To further confirm the impact of this charge interaction, sodium lauryl sulfate, an anionic surfactant widely used in pharmaceutical formulations [18], was employed due to its negative charge, which contrasts with that of chloroquine. Both of these substances showed their capability to influence the zeta potential of artificial chylomicron particles in preliminary experiments. Using the previously developed in vitro model, this study investigated how the addition of these agents to the artificial chylomicron compartment could affect the uptake of model drugs into this compartment.

## 2. Materials and Methods

### 2.1. Materials

Rifampicin (≤100%, CAS: 557303) was procured from EMD Millipore Corp, Burlington, MA, USA, while quercetin (≥95%, CAS: 117-39-5), 1-octanol (99%, CAS: 111-87-5), and chloroquine as diphosphate salt (98.5–101.0%, CAS:5 0-63-5) were sourced from Sigma–Aldrich Co. (Saint Louis, MO, USA). Intralipid^®^ (20%) was obtained from Fresenius Kabi (Toronto, ON, Canada). Peanut, olive, and sesame oil products were acquired from a local Edmonton grocery, while coconut oil (CAS: 8001-31-8) was obtained from Medisca (Saint-Laurent, QC, Canada), and sodium lauryl sulphate (≤100%, CAS: 151-21-3) was obtained from Caledon Laboratories (Toronto, ON, Canada). Labrafil^®^ M 2125 CS was obtained from Gattefossé (Toronto, ON, Canada) while cannflavin (≥98%, CAS: 76735-57-4) was obtained from Cayman Chemical (Ann Arbor, MI, USA). Additionally, synthetic hydrophobic polyvinylidene fluoride (PVDF) membranes were acquired from Millipore affiliated with Merck KGaA, Darmstadt, Germany. For HPLC analysis, methanol (99.9%, CAS: 67-56-1) and acetic acid (≥99.7%, CAS: 64-19-7) of HPLC grade were obtained from Fisher Scientific (Ottawa, ON, Canada); all other reagents were of analytical grade.

### 2.2. Methods

#### 2.2.1. Franz Cell for Studying Intestinal Lymphatic Uptake

The receiver compartment of a Franz cell was filled with either Intralipid^®^ (20%) alone or Intralipid^®^ mixed with a potential enhancer or inhibitor, totaling 12 mL. Olive, sesame, peanut, and coconut oils were added at a 2% concentration to the Intralipid^®^ in the receiver compartment to explore their potential as enhancers for uptake in the model. Additionally, 5% (+/−) chloroquine and 2%, 1%, and 0.5% sodium lauryl sulfate were introduced into the receiver compartment containing Intralipid^®^ to investigate their impact on the zeta potential. The experimental setup was maintained at a temperature of 37.0 ± 0.5 °C, and magnetic stirring at 600 rpm was employed for fluid agitation. A hydrophobic PVDF membrane impregnated with n-octanol and with a pore size of 0.22 µm was positioned between the compartments. Within the donor compartment, 2 mL solutions (1 mg/mL) of the model drugs, rifampicin, quercetin, in methanol, and dimethyl sulfoxide, were introduced. In various experiments, the receiver compartment contained either Intralipid^®^ or a mixture of Intralipid^®^ with an enhancer or an inhibitor at specific percentages. Sampling was conducted at various time intervals (0–4 h), involving the withdrawal of 0.2 mL samples. A similar procedure was followed with cannflavin A in the donor compartment and Labrafil^®^ M 2125 CS added to the Intralipid^®^ in the receiver compartment to investigate the effect of adding Labrafil^®^ to the cannflavin formulation. Samples taken from the receiver compartment were subsequently extracted and subjected to an analysis of their drug content using a Shimadzu HPLC device (LC-10AD, Shimadzu Corporation, Kyoto, Japan) equipped with an SIL-10A (Shimadzu Auto Injector) and a UV–VIS detector (SPD-10AV). The analysis was performed via a Kinetex^™^ C18 column (250 mm × 4.6 mm, i.d.—5 μm) from Phenomenex (Torrance, CA, USA) [9]. The column temperature was maintained at 25 °C, and specific analysis conditions for each drug are listed in Table 1. The resulting peak areas were integrated using LabSolutions software (Version 5.3, Shimadzu Corporation, Kyoto, Japan). Cannflavin A was detected using the method developed by O’Croinin et al. [19]. This methodology utilizes electrospray ionization liquid chromatography–mass spectrometry (LC-MS) analysis to separate and quantify cannflavins using an efficient isocratic elution. The LC-MS system consisted of a Phenomenex Luna^®^ 3 µm C18 (2) 100 Å 150 × 4.6 mm (Torrance, CA, USA) column for separation and a single-quadrupole mass spectrometry apparatus (Shimadzu, Kyoto, Japan) for the quantification of cannflavin A in a positive single-ion monitoring mode.

#### 2.2.2. Measurement of Zeta Potential of Intralipid^®^

Zeta potential measurements of both Intralipid^®^ and the Intralipid^®^ mixtures with different agents, specifically (+/−) chloroquine at 10%, 5%, 2.5%, 1.25%, and 0.125% in addition to sodium lauryl sulfate at 2%, 1%, and 0.5%, were conducted using dynamic light scattering, employing a Malvern Ultra Zeta Sizer (Malvern, United Kingdom). This analysis was conducted at a temperature of 25 °C, using polystyrene latex cells (DTS0012) in triplicate for each sample. The obtained results were analyzed using Malvern Panalytical software (version: 2.1.0.15).

#### 2.2.3. Statistical Analysis

Statistical analysis was conducted using GraphPad Prism software version 10.10.3 (GraphPad Software, San Diego, CA, USA). For comparisons between two groups, paired *t*-tests were performed, or one-tailed *p*-values were determined. A significance level of α = 0.05 was applied, and in all cases, *p*-values of less than 0.05 were considered indicative of statistical significance.

## 3. Results and Discussion

### 3.1. Effects of Different Oils Augmenting In Vitro Intestinal Lymphatic Uptake

Chylomicrons, as lipoproteins rich in triglycerides, have triglycerides primarily derived from dietary sources [8]. This emphasizes the significance of diet and lipid-based prodrugs and formulations in promoting the production of chylomicrons [20], which would consequently promote the uptake of drugs delivered via such formulations and delivery systems into the intestinal lymphatic system through the chylomicron pathway, ultimately enhancing the bioavailability of potential therapeutic agents [21,22].

In the quest for these effects, the utilization of oils containing long-chain fatty acids emerged as a prominent strategy. Long-chain triglycerides are the primary constituents of chylomicrons [8,23]; therefore, oils rich in long-chain fatty acids, such as sesame oil, olive oil, and peanut oil, are frequently employed [15,24,25]. Typically, long-chain triglycerides undergo re-esterification and become part of chylomicrons, allowing them to enter the lymphatic system. In contrast, medium-chain triglycerides are known to be transported mainly via the portal pathway [25,26]. Few studies have reported that medium-chain triglycerides appear in human chylomicrons after their oral administration [15,27]. In this study, coconut oil, distinguishable by its high proportion of medium-chain triglycerides fatty acids [28], was incorporated to enhance the breadth of the comparison with other oils. All other oils are abundant in long-chain fatty acids, which are recognized for their lymphatic transportation properties. In the assessment of the potential of various oils to enhance intestinal lymphatic uptake, as depicted in Figure 1, distinct patterns of uptake emerged. Coconut oil showcased the most prominent early-stage release effect. Olive oil initially exhibited a lesser magnitude of uptake enhancement than coconut oil, yet toward the later stages of the evaluation, it had a similar enhancing effect, resulting in a 3.5-fold increase in uptake. Moreover, sesame oil initially exhibited a similar enhancement pattern to that of olive oil. However, its release profile eventually matched that of peanut oil, which demonstrated the least pronounced impact. As previously reported, the addition of peanut oil to Intralipid^®^ resulted in a 1.5-fold increase in the lymphatic uptake of rifampicin in the in vitro model [9].

The fatty acid compositions of the diverse oils used in this work are detailed in Table 2. Coconut oil distinguishes itself with an abundance of saturated medium-chain fatty acids, setting it apart from other oils that are predominantly composed of long-chain fatty acids [28,29]. This specific characteristic renders it the least likely, if at all, to impact lymphatic uptake in vivo [30,31]. However, it also corresponds with the performance of coconut oil when compared to its counterparts in vitro. Its superiority in enhancing uptake in the in vitro model can be traced to its elevated content of medium-chain fatty acids, which exhibit heightened water solubility in comparison with the other oils [28,32]. This increased water solubility is thought to facilitate the integration of the oil into the aqueous external phase of Intralipid^®^. Therefore, coconut oil served as a favorable vehicle for the drug, facilitating its capture within the artificial chylomicron particles within the receiver compartment of the used model.

While medium-chain triglycerides are not the primary choice for facilitating intestinal lymphatic transport, they do play a significant role in minimizing fluctuations in drug absorption through the lymphatic route [33]. When included in lipid formulations designed to promote enhanced intestinal lymphatic uptake, medium-chain triglycerides—when combined with natural oils containing long-chain triglycerides—have demonstrated the ability to improve drug emulsification and the micellar solubilization of the tested lipid-based formulation [29]. This, in turn, resulted in a more consistent drug concentration within the lymphatic system. Nevertheless, it is worth mentioning that while the addition of medium-chain triglycerides reduced variability [34], various formulations with only natural oil vehicles containing long-chain fatty acids still outperformed others in terms of both lymphatic transport and systemic in vivo bioavailability [33,34].

The effects of the remaining oils, namely olive oil, sesame oil, and peanut oil, aligned with their in vivo behavior in enhancing lymphatic uptake. These oils share a high content of long-chain fatty acids [20,21,22], including stearic acid (C18:0), oleic acid (C18:1), linoleic acid (C18:2), and palmitic acid (C16:0), which are major fatty acids found in chylomicrons at varying proportions [25]. Yet the degree and type of unsaturation within these fatty acids arise as factors determining their effectiveness in enhancing intestinal uptake. The existing literature emphasizes that vegetable oils containing higher levels of oleic acid and linoleic acid tend to have a more favorable impact on promoting intestinal lymphatic absorption [35,36]. Olive oil and sesame oil, in particular, are abundant in these unsaturated C18 fatty acids [37,38] and have been documented to enhance both intestinal lymphatic and systemic transport more effectively than other vegetable oils [15]. 

Similar to what has been reported previously, the findings of this study demonstrate that olive oil performed better than sesame oil in facilitating the transportation of drugs into the receiver compartment. However, it is essential to acknowledge that sesame oil can display variable performance due to its susceptibility to oxidation, a significant concern for oils, especially those rich in linoleic acid [39]. Sesame oil, in comparison to olive oil, contains a higher proportion of linoleic acid (Table 2). Consequently, the observed differences in performance between sesame oil and olive oil may be attributed to a decline in linoleic acid levels caused by potential oxidation mechanisms. Similarly, peanut oil aligned with its previously reported in vivo performance, which identified it as the least effective in enhancing lymphatic transport among oils containing long-chain triglycerides [15]. The reported results may be attributed to the specific composition of this oil as it contains the lowest percentage of fatty acids known to enhance chylomicron lymphatic transport [40].

**Table 2 pharmaceutics-16-00768-t002:** Fatty acid compositions of the oils investigated to enhance the lymphatic uptake of rifampicin in the developed model.

Fatty Acid	Length: Saturation	% of Fatty Acid in Different Oils			
		Coconut Oil [28]	Olive Oil [37]	Sesame Oil [38]	Peanut Oil [40]
Capric Acid	C8:0	7	-	-	-
Caprylic Acid	C10:0	8	-	-	-
Lauric Acid	C12:0	49	-	-	-
Myristic Acid	C14:0	8	-	-	-
Palmitic Acid	C16:0	8	7.5–20	11–16	11–14
Stearic Acid	C18:0	2	0.5–5	11–16	-
Oleic Acid	C18:1	6	55–83	35–46	45–53
Linoleic Acid	C18:2	2	3.5–21	40–48	27–32
Linolenic Acid	C18:3	-	-	0.5	-
Arachidic Acid	C20:0	-	-	-	1–2
Behenic Acid	C22:0	-	-	-	1.5–4.5

As explained in our previous study [9], incorporating lipolysis or another digestion model into the developed model could address the complexities of the gastrointestinal journey for various fatty acids and co-administered drugs [15,41]. Such an approach would help estimate the implications of different digestion processes for the absorption of co-administered drugs and, coupled with this model, may offer additional insight into their potential lymphatic uptake when applicable.

Various lipid-based formulation excipients and drug delivery systems have been and continue to be developed to deliver drugs through intestinal lymphatics [1,7,21]. To assess the suitability of our in vitro model for formulation development, we examined the uptake of cannflavin A with and without Labrafil^®^. The results presented in Figure 2 demonstrate the efficacy of the model in studying the impact of formulation excipients on intestinal lymphatic uptake, with Labrafil^®^ enhancing cannflavin uptake.

### 3.2. The Effect of Changing the Zeta Potential of Artificial Chylomicrons on Lymphatic Uptake

(+/−) Chloroquine is a drug that is primarily employed for malaria prevention and treatment [42]. This compound possesses dibasic characteristics, featuring two basic groups corresponding to the nitrogen in the quinoline ring and the diethylamino side-chain nitrogen. These groups possess ionization constants of 8.1 and 10.2, respectively [43]. At physiological pH levels of around 7.4, (+/−) chloroquine predominantly undergoes ionization in its mono-protonated form, while in lower-pH regions of the body, it can transition into its di-protonated state (Figure 3) [17].

As illustrated in Figure 4, the introduction of 5% (+/−) chloroquine into the Intralipid^®^ within the receiver compartment led to a reduction in drug release for both rifampicin and quercetin. Specifically, when (+/−) chloroquine was added, only a mere 0.4% of the release achieved without (+/−) chloroquine was observed for rifampicin. Similarly, with quercetin, the presence of (+/−) chloroquine resulted in approximately 1% of the release that was documented in its absence. 

This inhibition mechanism is assumed to arise from the positively charged nature of the (+/−) chloroquine within the donor compartment. This positive charge might have prompted an interaction with the negatively charged Intralipid^®^ particles, thus impeding the entry of the tested drugs into the artificial chylomicron particles. To validate this hypothesis, zeta potential measurements were conducted for Intralipid^®^ both with and without the addition of (+/−) chloroquine. From Figure 5, it is evident that as the percentage of (+/−) chloroquine increases, a corresponding rise in the neutralization of the negative charge on Intralipid^®^ occurs. This trend resulted in a reduction in the zeta potential on the Intralipid^®^ particles.

In our previous publication, we demonstrated how a PL-81 coating encapsulated Intralipid^®^ particles, thus hindering drug penetration [9]. This confirmation was supported by microscopic images, thus raising the question of whether there might be another biophysical mechanism for PL-81 chylomicron blockage. In this context, the primary objectives are two-fold: first, to determine if the inhibition mechanism of chloroquine relies solely on the presence of the coating, and second, to investigate the potential involvement of the zeta potential in this process.

To address the second part, an alternative agent was introduced with the aim of elevating the zeta potential of the Intralipid^®^ particles. This part of the experiment aimed to investigate whether enhancing the zeta potential would translate to an increased in vitro lymphatic drug uptake or not. Thus, sodium lauryl sulphate (SLS) was used to increase the zeta potential to see if that would increase the lymphatic uptake via the model used.

Sodium lauryl sulfate (SLS) is an alkaline, anionic surfactant with versatile applications. Within pharmaceutical formulations, SLS fulfils various roles, including those of an emulsifying agent, modified-release facilitator, penetration enhancer, solubilizing agent, tablet, and capsule lubricant [18,44]. Upon incorporating SLS into the Intralipid^®^ within a concentration range of 0.5–2%, an escalation in the zeta potential was observed, as illustrated in Figure 6. Investigating the impact of varying SLS percentages on the release of rifampicin revealed intriguing insights. At a 0.5% SLS concentration, there was almost no change in the zeta potential, and the effect on drug release remained minimal as well. The addition of 1% and 2% SLS translated into zeta potential increases of approximately 1.4 and 1.6 times, respectively. Correspondingly, these SLS levels yielded enhancements of 1.6 and 1.2 times the uptake within the in vitro model. 

The literature indicates that SLS may enhance absorption, possibly through a connection with the cAMP system. In this current experimental setup, as there is no cAMP system involved, it is suggested that the effect can be physicochemical rather than biological [18]. The experiment aimed to test the hypothesis that increasing the zeta potential through the addition of sodium lauryl sulfate (SLS) would enhance uptake in the in vitro model. Still, it is important to note that SLS functions as an anionic emulsifier within a concentration range of 0.5–2.5% [45]. Therefore, another potential explanation for the observed results could be provided by the emulsifying role of sodium lauryl sulfate. At lower concentrations (0.5% and 1%), SLS would be adsorbed at the oil–water interface, facilitating the uptake of rifampicin into the internal phase droplets. However, as the concentration increased (2%), the interface could have become saturated, indicating an excess of SLS molecules covering the available surface area. Consequently, once the interface reached saturation, the uptake of rifampicin into the droplets would become more challenging. As a result, little difference was observed in the latter case (2% SLS) compared to the scenario in which no SLS was added to the medium.

Yet if coating was only the factor affecting uptake, SLS would have decreased it similar to PL-81, which was found to coat the Intralipid^®^ particles and hence impede rifampicin uptake. The outcomes acquired from this study may potentially imply the existence of an optimal concentration range of sodium lauryl sulphate wherein the uptake enhancement becomes apparent. Nonetheless, these findings underscore the affirmative influence of a zeta potential increase on uptake within the in vitro model. Moreover, the precise mechanism through which alterations in the zeta potential produce these uptake effects necessitates further comprehensive investigation.

## 4. Conclusions

In this study, our previously developed in vitro model was utilized to further investigate how various agents influence drug uptake into artificial chylomicrons (Intralipid^®^). Typically, long-chain fatty acids facilitate intestinal lymphatic uptake, while medium-chain counterparts are mainly absorbed through the portal pathway. The results showcased the ability of the model to distinguish between oils containing long-chain fatty acids, particularly olive, sesame, and peanut oils, yet it did not capture the difference between these long-chain rich oils and a medium-chain-rich oil (coconut oil) in terms of lymphatic uptake. The increased uptake observed with coconut oil was attributed to its better emulsification in an artificial chylomicron medium due to its composition of medium-chain fatty acids. Moreover, the enhanced uptake of the tested formulation with linoleoyl polyoxyl-6 glycerides emphasized the practical utility of our model in optimizing formulations. Additionally, the findings indicated that adjusting the zeta potential, increasing it using sodium lauryl sulfate (SLS) and decreasing it using (+/−) chloroquine, resulted in corresponding increases and decreases in uptake in the in vitro model. These results underscored the potential influence of the zeta potential on intestinal lymphatic uptake in our model. Nevertheless, further research is necessary to explore whether this mechanism holds true in vivo.

## Figures and Tables

**Figure 1 pharmaceutics-16-00768-f001:**
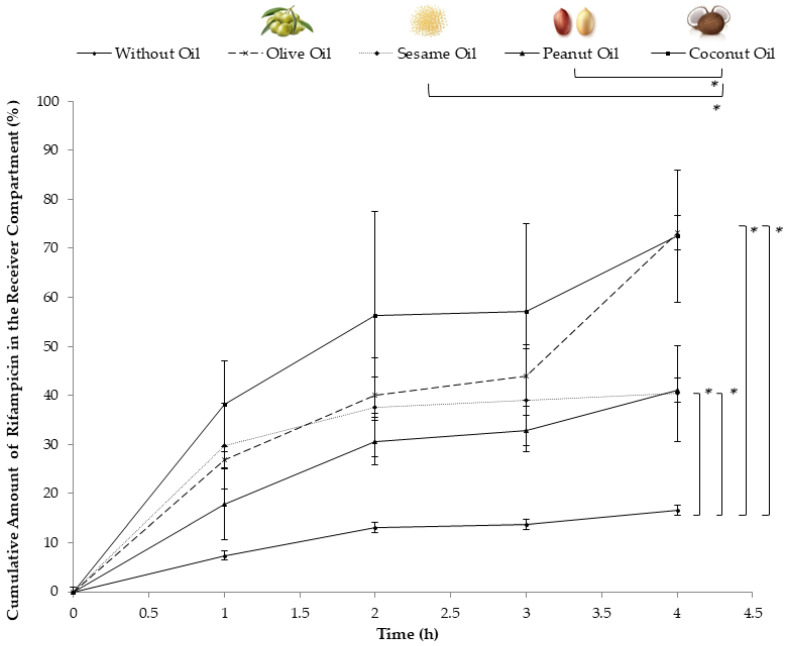
A demonstration of the increased in vitro lymphatic uptake of rifampicin via the developed model when 2% concentrations of different oils were added to Intralipid^®^. Data represent mean ± SE values (*n* = 3). * Indicates statistical significance (*p* < 0.05) between different groups. Images of different components were obtained from designers via Freepik.com.

**Figure 2 pharmaceutics-16-00768-f002:**
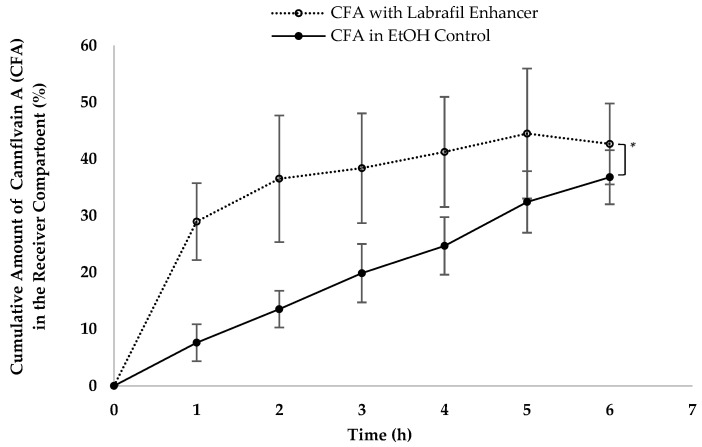
A demonstration of the increased in vitro lymphatic uptake of cannflavin A (CFA) via the developed model when Labrafil^®^ 2125 CS (an uptake enhancer) was added to Intralipid^®^. Data represent mean ± SD values (*n* = 6). * indicates the statistical significance (*p* < 0.05) between the different groups.

**Figure 3 pharmaceutics-16-00768-f003:**
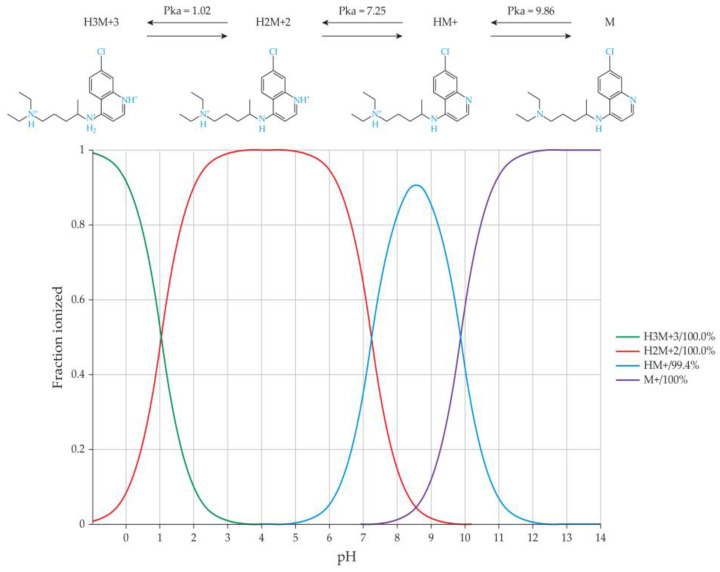
Microspecies of chloroquine at different pH values (0–14) demonstrating the ionization behaviour of chloroquine throughout this range of pH values.

**Figure 4 pharmaceutics-16-00768-f004:**
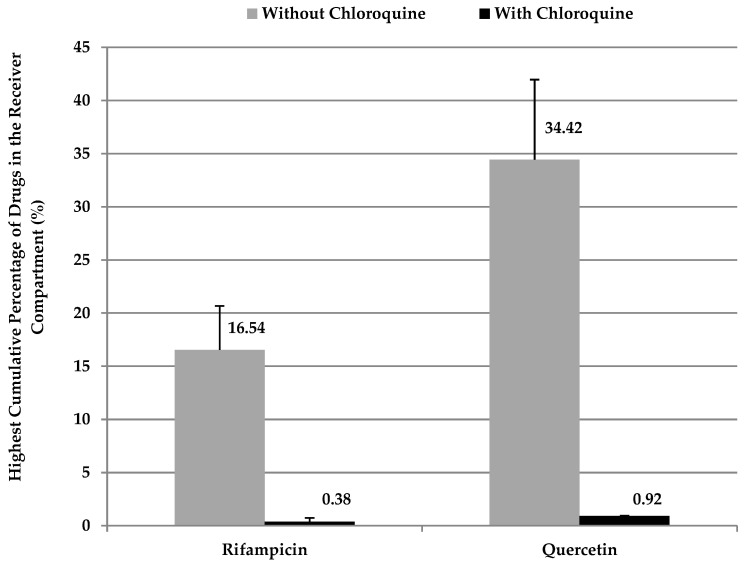
An illustration of the differences in the percentage of the in vitro lymphatic uptake of the model drugs, rifampicin (16.54 ± 4.13) and quercetin (34.42 ± 7.53), via the developed model when 5% (+/−) chloroquine was added to the Intralipid^®^ in the receiver compartment of the model. Upon performing this action, the uptake decreased to (0.38 ± 0.35, *p* < 0.05) and (0.92 ± 0.01, *p* < 0.05) for rifampicin and quercetin, respectively.

**Figure 5 pharmaceutics-16-00768-f005:**
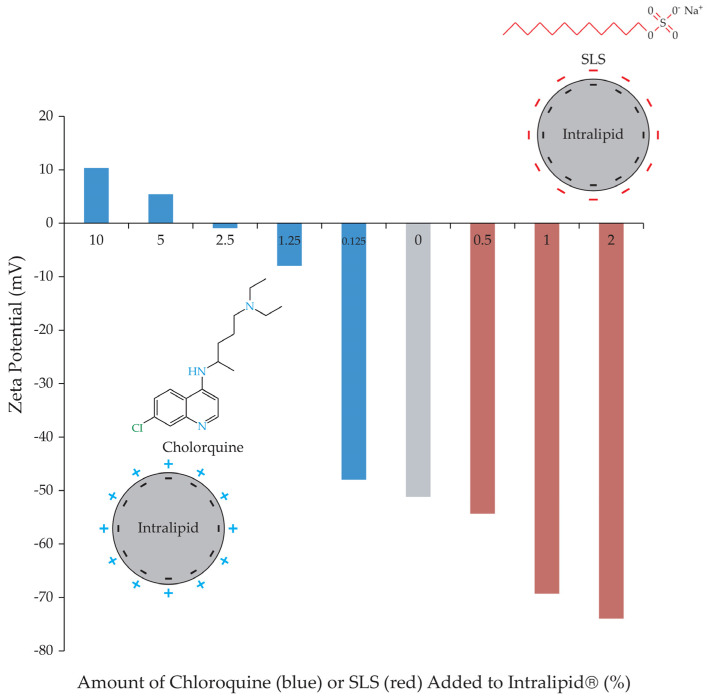
Change in Intralipid^®^ zeta potential with varying percentages of chloroquine (blue) and sodium lauryl sulphate (SLS, red), expressed as average of triplicates. Zeta potential of Intralipid^®^ alone is shown in grey.

**Figure 6 pharmaceutics-16-00768-f006:**
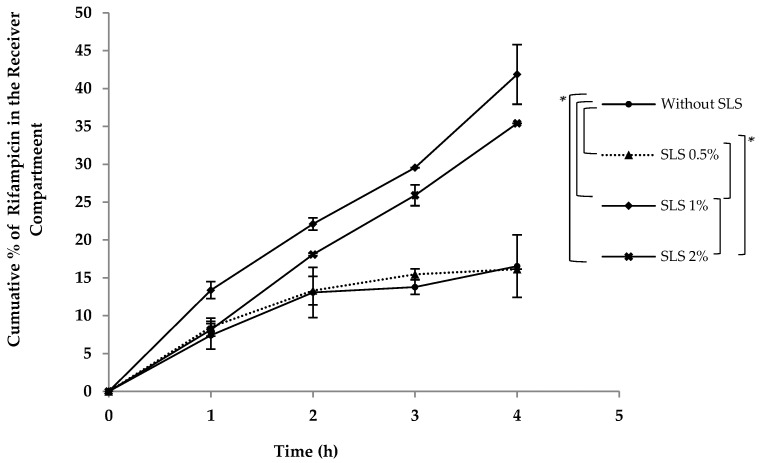
A demonstration of the increased in vitro lymphatic uptake of rifampicin via the developed model when different percentages of sodium lauryl sulphate (SLS) were added to Intralipid^®^ in the receiver compartment. * Indicates statistical significance (*p* < 0.05) between different groups.

**Table 1 pharmaceutics-16-00768-t001:** HPLC analysis conditions for model lymphotropic drugs.

Model Drug	Mobile Phase	Flow Rate (mL/min)	Detection Wavelength (nm)
Rifampicin	Methanol–Acetate Buffer (pH = 5.8) (60:40)	1.2	254
Quercetin	Methanol–Acetate Buffer (pH = 5.8) (60:40)	1.2	257, 370

## Data Availability

Data are contained within the article.

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
