# Peer review of "In Vitro Predictive Model for Intestinal Lymphatic Uptake: Exploration of Additional Enhancers and Inhibitors"

_pharmaceutics, 2024, doi:10.3390/pharmaceutics16060768_

Round 1

Reviewer 1 Report

Comments and Suggestions for Authors

This article ‘In Vitro Predictive Model for Intestinal Lymphatic Uptake: Exploration of Additional Enhancers and Inhibitors’ introduces the investigation of the influence of the emulsion particles zeta potential and some different oils to the intestinal lymphatic uptake. This article has certain practical significance and scientific nature. However, this article has not enough data to match the 《Pharmaceutics》.

Comments on the Quality of English Language

Good.

Author Response

Attached, thanks!

Reviewer 2 Report

Comments and Suggestions for Authors

Yousef et al.'s research applied their previously established in-vitro compartmental model that forecasts drug partitioning into chylomicrons under various oil formulations. Through the comparison, they identified that coconut oil surpassed other oils facilitating the drug partitioning into chylomicrons. In addition, they also identify the zeta potential as important factor influencing the drug penetration into chylomicrons. I have several questions that need clarification from the authors before the paper can be published.

1.     The authors used the Rifampicin and quercetin as the tool drugs to investigate the chylomicrons partitioning and intestinal lymphatic absorption. However, these two drugs are not subject to very high chylomicrons partitioning in general. Does the author consider more lipophilic drugs, ie. Log D >5 (probucol, and Halofantrine, etc), to confirm if the findings in Rifampicin and quercetin can be translatable to these high lipophilic drugs.

2.     I am not how the in-vitro model work, particularly for receiver compartment which contains the acritical chylomicrons. May I ask if the oil formulation will be digested and form as artificial chylomicrons in the receiver compartment, or the acritical chylomicrons  already pre-exist, and the oil formulation just facilitate the drug distribute to the artificial chylomicrons. Please make more clarity to the reader for the system setting.

3.     Line 21, “due to its composition”, please provide more clarity.

4.     Line 60-61, please explain why the drug release needs to be suppressed in the in-vitro model. Does that aim at “artificially” boost the chylomicrons partitioning?

5.     Line 75-76. Please clarify what’s the racemic chloroquine for assay purpose, and what’s sodium lauryl sulfate for assay purpose. Just stating these two as pharmaceutical ingredients and excipients doesn’t provide any meaning

6.     Please revise the all figure Y axis as the legend text and number scale overlapping. 

Comments on the Quality of English Language

Suggest edits throughout the text to improve the readability

Author Response

Kindly find it attached, thanks!

Reviewer 3 Report

Comments and Suggestions for Authors

In this manuscript, the authors used an in vitro model to study the effects of potential enhancers or inhibitors on the lymphatic absorption of lipophilic drugs. Although the results are relevant for a better understanding of drug absorption via lymphatic system, control experiments are missing, especially for results shown in Figure 1. The authors discussed potential different effects of medium-chain fatty acids and long-chain fatty acids, a control experiment comparing pure medium-chain fatty acids and long-chain fatty acids with the natural oils could be more convincing.

Two minor issues:

Line 159: " In contrast, olive oil initially exhibited a similar magnitude of uptake enhancement, ..." It is not clear what is olive oil compared to here, coconut oil or sesame oil? From the context, the comparison refers probably to coconut, in this case, the magnitude of enhancement was obviously lower in the early phase.

Line 407: "At 0.5% SLS concentration, the zeta potential nearly doubled;". If this refers to Figure 5, there was almost no change of zeta potential with 0.5% SLS.

Author Response

Please find the responses in the attached document, thanks!

Reviewer 4 Report

Comments and Suggestions for Authors

This paper is a follow on from previous work. In the current version it is almost impossible to read this paper without having the previous paper to hand which means that significant revisions are required. 

The presentation of the results and discussion in this paper needs to be improved. The results are lost in an extensive dialogue that include elements of introduction and discussion. It should have a lot more focus and clearly describe firstly the data generated then the implications of this finding.

Specific revisions:

Line 85 check rifampicin purity as >100%?

Table 2 belongs in materials and methods and not results (it does not contain any of your data)

The y axis on the figures cannot be read (this may be due to the line numbers but as a reviewer it is hard to read)

Line 212-224 are neither results nor discussion - this is making the results hard to read - this is throughout the section

Your results are limited to an in vitro model and you have not correlated to in vivo data. Your discussion therefore needs to be limited to your own data - it currently extends way beyond what you have shown

Line 267 includes results on Labrafil but this is not included in your methods. Please add a method for this section or delete the results

Section 3.2 is better as it discusses your results however lines 380-388 are not related to your results so could be deleted

In the section on addition of SLS there is no discussion on the solubilising capacity as a mechanism driving absorption via a concentration gradient

I think that the abstract and conclusion need to be clearer on reporting that the model only replicates rank order for long chain fatty acids and that it is not valid for medium chain fatty acids

Both the abstract and conclusion also need to be focussed on the data within this study and not beyond.

Author Response

Kindly find the responses in the attached document. Thank you for your time.

Round 2

Reviewer 2 Report

Comments and Suggestions for Authors

The author has addressed my comments. I have no other comment. 

Comments on the Quality of English Language

None

Reviewer 3 Report

Comments and Suggestions for Authors

The revised version can be accepted.